



# The influence of sea surface temperature on the intensity and associated storm surge of tropical cyclone Yasi: A sensitivity study

Sally L. Lavender, Ron K. Hoeke and Deborah J. Abbs

Climate Science Centre, CSIRO Oceans and Atmosphere, PMB1, Aspendale, VIC, 3195, Australia

*Correspondence to*: Sally L Lavender (Sally.Lavender@csiro.au)

**Abstract.** Tropical cyclones (TCs) cause widespread damage associated with strong winds, heavy rainfall and storm surge. Understanding changes in these characteristics associated with potential future climate scenario sea surface temperatures (SSTs), as well as variations with climate modes, such as the El Niño/Southern Oscillation, is important for mitigating impacts. TC Yasi was one of the most powerful TCs to impact the Queensland coast

since records began. Prior to Yasi, the SSTs in the Coral Sea were higher than average by 1-2°C, primarily due to the 2010/2011 La Niña event. In this study, a conceptually simple sensitivity analysis is performed to gain insight into the influence of SST on the track, size, intensity and potential destructiveness of TC Yasi, including rainfall and storm surge.

In order to assess the ability of a high resolution regional model at simulating TC Yasi, the Weather Research and

Forecasting (WRF) model is forced in a control run using atmospheric reanalyses and observed SST data over the period 31$^{st}$ January to 4$^{th}$ February 2013. The model is able to closely simulate the observed track, with the modelled landfall occurring within 50 km and 3-hours of the observed event. Additional simulations are carried out with uniform SST anomalies of between -4 °C and 4 °C applied to the observed SST's over the whole region in 1 degree increments, forming a set of nine simulations. The resulting surface winds and pressure were then used to force a

barotropic storm surge model.

An increase in SST results in an increase in intensity, precipitation and destructiveness of the storm, however there is little influence on track prior to landfall. In addition to an increase in precipitation, there is a change in the spatial distribution of precipitation as the SST increases. Decreases in SSTs result in an increase in the radius of maximum winds due to an increase in the asymmetry of the storm, although the radius of gale-force winds decreases. These

changes in the TC characteristics also lead to changes in the associated storm surge. Generally, cooler (warmer) SST lead to reduced (enhanced) maximum storm surges. However, the increase in surge reaches a maximum with an increase in SST of 2 °C. Any further increase in SST does not affect the maximum surge but the total area and duration of the simulated surge increases with increasing upper ocean temps. The largest change in storm surge



occurs when a negative SST anomaly is applied with a decrease in storm surge height of over 3m when the SST is reduced by 2 °C.

In summary, increases in SST lead to an increase in the potential destructiveness of TCs, although this relationship is not linear.

# 1 Introduction

Tropical cyclone Yasi was one of the most powerful tropical cyclones (TCs) to impact the Queensland coast since records began. Yasi made landfall in the early morning of 3rd February 2011, near Mission Beach, as a Category 5 cyclone. Prior to Yasi, the SSTs in the Coral Sea were higher than average by 1-2°C due to a La Niña event (e.g. Ummenhofer et al., 2015). A large storm surge was associated with TC Yasi, although it made landfall at low tide, with a 5 m surge observed in Cardwell which is 2.3 m above the highest astronomical tide (BoM, 2011). This study investigates what impact these higher than usual SSTs might have had on the track, intensity and size of TC Yasi and changes to the (non-tidal) associated storm surge.

The importance of warm SSTs for the development and intensification of TCs has been long known: surface fluxes of latent and sensible heat from the oceans provide the potential energy to TCs (Ooyama 1969; Emanuel 1986). Palmén (1948) was the first to document that TCs only occur over oceans warmer than a critical temperature of 26—27°C and subsequently the values of 26°C and 26.5°C have been widely used throughout TC research (e.g. Gray 1968; Holland 1997) as a threshold SST for the formation of TCs. This threshold temperature was recently revisited by Dare and McBride (2011) using observations from 1981 to 2008 with results consistent with these earlier studies. They found that the majority (93%) of TCs occur at SSTs greater than 26.5°C and over 98% at SSTs greater than 25.5°C. The positive trend in SSTs over their study period has not lead to a shift in this threshold temperature.

Although SSTs are clearly an important factor to consider when examining TCs, recent research has suggested there should be less importance placed on SSTs alone and more on the surface fluxes and wind speed that are the drivers of the energy of the TCs (Emanuel 2007). This is of particular importance when considering how TCs may change in a warmer world. The initial assumption was that an expansion of the extent and duration of ocean areas above the 26 °C SST threshold in future will lead to the formation of more TCs. However, this has been found to not necessarily be the case, with many studies projecting a decrease in TC activity in a warmer world (e.g. Knutson et al. 2010). It is the relative SST (i.e. local SST relative to the global tropical mean), rather than absolute SST,



that has been found to be important in determining changes in frequency and intensity of TCs (e.g. Ramsay & Sobel 2011) resulting in different projections depending on the region (Vecchi & Soden 2007).

Idealised SST sensitivity experiments are not new. Evans et al. (1994) used a regional model to impose SST anomalies and analyse changes in 2 Australian TCs, showing potential for the intensity of TCs to increase as SSTs

increase whilst also pointing out the caveats of an idealised study such as the present one. More recently, Miglietta (2011) examined the influence of uniform SST anomalies on "medicanes" (TC-like cyclones in the Mediterranean) finding that when SST is reduced by more than 4°C the cyclone loses tropical cyclone like characteristics. Kilic and Rable (2013) also use SST sensitivity experiments to confirm the linear relationship between SST and TC intensity. The present study uses a similar methodology to highlight the influence of SST when all other variables

remain the same, which is not what we could expect in the real-world under climate change. However, it allows us to examine the sensitivity of TCs to SST changes alone in a way that would be difficult with real-world observations. A larger focus on precipitation and the influence on storm surge adds an additional new dimension to this work from previous studies.

Parker et al. (2017) examined the influence of atmospheric and SST intitaliasation data as well as the choice of

parametrisation schemes on the track, intensity, landfall location and intensity and translational speed on simulations of TC Yasi. They found the choice of cumulus parametisation made the biggest difference with a trade-off needed between accurate trajectory and more realistic intensities.

TC Yasi occurred during a season when SSTs over the south-west Pacific region remained above average (Imielska 2011). The fact that this was the most powerful TC to affect Queensland in over 90 years leads to the question of

how this storm may have been influenced by these higher than average SSTs. This may also provide insights of how we might expect the potential destructiveness of TCs to change in a warmer climate. Here we perform a conceptually simple sensitivity study using limited area models to analyse the influence of SST on TC Yasi.

The following section describes the numerical modelling setup and the data used to initialise the model. Section 3 evaluates the ability of the model to simulate TC Yasi. The sensitivity of the track, intensity, precipitation and

storm surge associated with TC Yasi is presented and discussed in Section 4. A summary is presented in Section 5 including a discussion of the limitations of this study.





## 2 Methodology

### 2.1 Data

Atmospheric and surface data required to initialise the model were obtained from ERA-Interim re-analysis data from the European Centre for Medium range Weather Forecasts (ECMWF; Dee et al. 2011) on a $1.5 \times 1.5$ grid for the period 28[th] January to 4[th] February 2011 as six-hourly data.

Daily sea surface temperature data from the real-time global SST analysis was obtained from the National Centers for Environmental Prediction/Marine Modeling and Analysis Branch (NCEP / MMAB) on a 0.5 degree grid over the same time period as above.

The observed track of TC Yasi, as well as central pressure and maximum windspeeds were obtained from the International Best Track Archive for Climate Stewardship dataset (IBTrACS; Knapp et al. 2010) which incorporates data from the Australian Bureau of Meteorology.

### 2.2 Model configuration

The Weather Research and Forecasting (WRF) model, Version 3.4 is used with a vortex-following, two-way nesting configuration. There are 3 domains. The outer grid has a horizontal resolution of 36 km. Both inner grids, 2 and 3, are able to move and have a grid spacing of 12 km and 4 km respectively. All grids have 36 vertical levels with a model top of 20 hPa. Only the outer grid is forced with the atmospheric and SST data. The following parameterisations were selected based on a number of tests and using previous analysis results: the Thompson et al. (2008) microphysics, the Rapid Radiative Transfer Model (RRTM) for longwave radiation (Mlawer et al. 1997), the Goddard shortwave scheme (Chou & Suarez 1994) and the Mellor Yamada Nakanishi and Niino Level 2.5 TKE scheme for the planetary boundary layer (Nakanishi & Niino 2006). The Kain-Fritsch (K-F; Kain 2004) cumulus parameterisation scheme was used on all grids.

Although the experimental design is conceptually simple, we make use of the simple mixed-ocean layer model capability in WRF due to the SST cooling that occurs after passage of a TC. This is an important feedback and one that should be included when making any observations based on SST changes. The initial mixed layer depth was set to 50 m and the temperature lapse rate below the mixed layer to 1.4 K m$^{-1}$.

The control run (CTRL) used the observed SSTs and a subsequent set of 8 simulations used these SSTs with an imposed temperature anomaly across the whole domain of -4°C, -3°C, -2°C, -1°C, 1°C, 2°C, 3°C and 4°C. All experiments were initialised at 00Z on 31[st] January 2011 and ran for 4 days. The only differences between the



experiments at initialisation were the SSTs. This design will also result in a change in the "relative" SSTs as well as absolute since SSTs in the global tropics are not changing, and are outside the domain.

The control run was repeated using both ERA-interim high resolution data (0.75 × 0.75 degree grid) and NCEP-FNL (1× 1 degree grid; NCEP 2000) data to force the model. Using the high resolution ERA-interim made no significant difference to the simulation. However, there were larger errors in the modelled track when NCEP-FNL data was used to initialise the storm. Using NCEP-FNL resulted in a smaller bias in the intensity, however the storm reached maximum intensity too quickly and started decaying before making landfall. The storm size in the simulations is too large (see Section 3) compared to observations, so we tested the effect of implanting a bogus vortex of different sizes. However, in terms of the track, the simulation without a bogus was more comparable to observations. An additional simulation with the convective parameterisation turned off on the inner grid was performed; however this resulted in a poorer representation of the intensity. Therefore, the control run was initialised using ERA-Interim 1.5 ° data, with no bogus vortex and convective parameterisation on all grids. Further limitations of this study will be outlined in Section 5.

## 2.3 Storm surge model

The Flow module of the Delft3D modelling system (open-source version 6.01.13.6455) was used to calculate storm surge (wind-setup and inverse barometer effect) along the Queensland coast. Delft3D Flow consists of a finite-difference solution to the Navier-Stokes equations for unsteady flow (Lesser et al., 2004); it was implemented using the shallow-water (depth-integrated) approximation on a coast-following curvilinear grid with grid resolution varying from approximately 5 km at the offshore boundary to approximately 1 km near the centre on the coast. Grid bathymetry and topography was generated from the ~250 m resolution Australian Bathymetry and Topography Grid (Whiteway, 2009) and wind and pressure fields were linearly interpolated from the WRF model output at 30-minute intervals and resulting (gridded) storm surge information stored at the same interval for all runs. Background sea levels were set to zero for all runs.

## 3  Simulation of Yasi in CTRL run

Figure 1a shows the track from the CTRL simulation (black) and that from observations (red). The model is able to closely simulate the correct track, with the modelled landfall occurring within 50 km and 3-hours of the observed event. The intensities of the simulated track are shown every 6 hours. At landfall the modelled TC is a Category 4 hurricane while the observations show Yasi reached Category 5 intensity by landfall. The intensities



(minimum pressure and maximum wind speed) are plotted against 6-hourly observations over the duration of the track in Fig 1b and c. At the CTRL simulation's initialization, the TC's vortex in the reanalysis dataset is too big and its intensity is too low, resulting in a difference in minimum pressure and maximum wind speed of 20 hPa and 12 m s$^{-1}$ respectively, at the start of the simulation. This bias of 20 hPa persists in the central pressure field (Fig 1b)

until after landfall when the simulated pressure remains too low. The modelled maximum wind speeds, however, follow the variability shown in the observations and at the maximum intensity the model only underestimates wind speed by 5 m s$^{-1}$. The wind speeds rapidly decrease after landfall and the slightly earlier landfall (by three hours) in the model relative to observations is clearly evident. After landfall, the model overestimates the maximum wind speeds by approximately 5 m s$^{-1}$. Parker et al, (2017) found large errors in the trajectory and landfall when using

the K-F scheme but more realistic intensities. These large errors in track are not evident in the current study.

The structure of the storm shortly before landfall in outgoing longwave radiation (OLR) and precipitation is compared to observations in Fig. 2. The difference in size due to the large vortex in the initial conditions is clearly evident; in particular the eye is too big. As mentioned in Section 2, the simulations were repeated with the initial vortex removed and a bogus vortex implemented in its place. However, in this case the storm became too small

and reached maximum intensity too quickly with the TC tracking to the south resulting in a large discrepancy in the track compared to observations. It is worth noting that Parker et al. (2017) found the K-F cumulus scheme resulted in a larger vortex than other schemes. Despite the large size of the simulated vortex, some of the small scale features in the OLR (Fig 2a and b) give confidence in the simulation, for example the cloud patterns in the southeast and northeast of the domain. Similarly, the precipitation (Fig 2c and d) show well defined rainbands,

with a large rainband swirling round to the northwest in both the model simulation and radar observations. The increased rainfall in the southeast as the system reaches the coast is also evident.

The CTRL simulation storm surge's overall extent (Fig 3a), duration and maximum non-tidal water levels storm surge of around 5 m appear to closely match (de-tided or residual) tide gauge observations (Fig 3b) and indications from debris lines (Queensland Government, 2012). However, consistent with the earlier and more southerly

landfall of the cyclone in the CTRL run, the timing of the simulated storm surge's peak occurs 2-3 hours earlier and further south (within Halifax Bay, red line in Figure 3b) compared to observations (near Cardwell, black line in Figure 3b). The influence of the magnitude of SST on the simulations of Yasi will now be analysed.



## 4 Influence of SST

The change in SSTs results in only subtle differences in the simulated tracks (Fig. 4a) with the largest differences occurring after landfall. Landfall timing and location remains similar in all simulations, with all runs making landfall within 1-degree (approx. 100 km) and 4-hours of the CTRL simulation. There is a small but not systematic change in landfall time in the experiments due to deviations in the tracks, with landfall occurring earlier by 1h30 for SST+1, 2h30 for SST+2 and 3h30 for SST+3 and SST+4. In the case of negative SSTs the largest difference occurs with SST-1 which results in landfall 2 hours after CTRL. The other negative SST anomalies only result in landfall 30-mins after the CTRL run. The sign of the SST anomaly has little influence on the latitude of landfall with the largest SST anomalies of both signs making landfall further south. However, after landfall the positive SST simulations have a tendency to move further southwards than the negative SST runs.

Differences in intensity between the SST experiments and the CTRL run are shown in Fig. 4b and c. The experiment setup means all simulations are initialised with the same pressure and wind fields. After 24 hours there are clear differences in the intensities, with larger differences occurring with larger temperature anomalies. The larger the positive anomaly the more intense is the storm with lower pressures and higher wind speeds. The larger the negative anomaly the opposite is true with lower intensities occurring. Increasing the SST has a larger influence on the minimum pressure than decreasing it with a maximum difference of -60 hPa occurring in SST+4 and 45 hPa in SST-4. The minimum pressure also occurs earlier in the run as the positive SST anomaly is increased. The earlier landfall is evident in the wind speed differences between the positive SST anomaly runs as the difference becomes negative when they make landfall prior to CTRL.

The radius of maximum winds (RMW) increases with cooler SSTs (Fig5a). This is consistent with the storm becoming less intense and more asymmetric. When the SST is increased by 1 degree, the RMW decreases, however, increasing the SST further doesn't decrease the RMW by much more. A more appropriate definition of the size of the storm is the radius of gale force winds ($> 17.5$ m s$^{-1}$; R17). R17 increases as SST increases (Fig 5b). The decrease in R17 with decreasing SST is much smaller. Although positive SSTs result in larger wind speeds at smaller radii (small RMW), the high wind speeds persist to larger radii.

The integrated kinetic energy (IKE, Powell & Reinhold 2007) takes into account both maximum wind speeds and storm size and is therefore a good measure of the destructiveness of a TC. Here we measure IKE for wind speeds greater than 17.5 ms$^{-1}$ (IKE$_{17}$) over the entire domain of grid 3. Increasing the SST anomaly from -4 to +4 shows a clear increase in the IKE$_{17}$ (Fig. 5c). This is due primarily to the increase in wind speeds (squared in the IKE calculation) as the SST anomaly increases.





Maximum overall storm surge heights in the SST experiments and the CTRL run are shown in Fig 6a. A large decrease in maximum storm surge corresponding to negative SST anomaly is evident, with a decrease in storm surge height of over 3m between the CTRL and SST-2 simulations. While maximum storm surge increases by about 1 m between the CTRL run and SST+1 and SST+2, further increases in SST anomaly do not lead to a further

increase in maximum storm surge. This may be due to a slight shift in the track southward and the TC approaching the coastline from a more northerly (less shore-normal) direction; it may also be due to localized changes in hydrodynamic momentum balances of individual bays and other coastal features, or some combination of the two. However, if the total area affected by storm surge within each simulation is calculated (Fig 6b), here arbitrarily defined as +1 m of water elevation, there is a clear and consistent increase in the area and duration of storm surge

from SST-4 to SST+4, qualitatively similar to the consistent increase in the TCs IKE.

The different characteristics between the simulated maximum storm surge and its timing and duration across the sensitivity runs are a reflection of storm surges sensitivity to cyclone forward speed, location of landfall and RMW and its complex interaction with local morphology (e.g. the shape and characteristics of coastal features such as bays and estuaries).

TCs cause widespread damage due to high wind speeds and storm surge. In addition, extreme precipitation events are also associated with the passage of TCs. Precipitation within 500 km of the TC centre in the control run reaches almost 6 mm hr$^{-1}$ km$^{-2}$ which is slightly higher than that recorded in TRMM 3B42 satellite data (not shown); this is expected due to the higher model resolution. The influence of SST on the precipitation is shown in Fig. 7a. Before landfall, the precipitation rate increases as both the SST increases and the intensity of the storm increases.

Increases in precipitation rate with increased positive SST anomalies are greater than the decreases with negative SST anomalies as there is a limit by how much the precipitation can decrease. After landfall, negative SST anomalies result in lower precipitation rates. Precipitation rates after landfall in the positive SST anomaly simulations are more variable, with only minor changes from the CTRL run.

The spatial distribution of rainfall shown as a 30-min rainfall rate for each of the 9 simulations at landfall is shown

in Fig 8. The increase in precipitation with increasing SST anomalies is clearly evident, leading to higher precipitation rates in both the inner and outer rainbands. A cooler upper ocean leads to a less distinct eye of the storm and increases the storm's asymmetry by elongating it in the meridional direction. The distribution of precipitation within the inner rainband shows a shift in position of the maximum rainfall from the front-left quadrant in SST-1 and CTRL to the front-right quadrant in SST+3 and SST+4. This is consistent with observations of TC-

rainfall (Lonfat et al. 2004). The increase in the size of the storm, in terms of the extent of its rainbands, as SST anomaly increases is also evident. The extent of the outer rainbands evident in Fig 8 also help to explain the



secondary peak in the area affected by storm surge (Fig 6b) when a greater than 1°C SST anomaly is applied since there will be high wind speeds associated with these.

In order to analyse the precipitation distribution in more detail, the precipitation was accumulated over different radii (Fig 7b; 0 - 100 km, 100 - 200 km, 200 – 300 km, 300 – 400 km and 400 – 500 km). The largest rainfall amounts occur between 100 km and 200 km from the centre of the storm in all simulations. This is in contrary to the maximum within 100 km shown in observations (e.g. Lonfat et al. 2004) and can be accounted for by the larger size of the eye relative to observations. A decrease in the SST leads to a decrease in the accumulated rainfall in all radii. However, increasing the SST from the CTRL value makes little difference to the precipitation rate within 100 km of the storm centre (red line Fig 7b). This is inconsistent with climate change studies which project an increase in the precipitation rate within 100 km (Knutson et al. 2010) but, as previously mentioned, is likely to be due to the larger vortex in the current study. The increase in SST results in a change in the distribution of rainfall with more falling in the 200 – 300 km band than within 100 km. It could be argued that Yasi was such an extraordinary TC in terms of size compared to other TCs impacting Queensland that it does not necessarily fit climate change studies.

## 5  Summary and discussion

This study uses an idealised numerical modelling sensitivity analysis to investigate the influence of imposed SST anomalies on TC Yasi. Yasi was a very powerful storm and occurred at a time when SSTs were above average; the atmospheric/mixed layer ocean model was able to correctly simulate the historical track and wind speeds (within 5 m/s), as well as small scale structures evident in the OLR and precipitation observations (the CTRL run, Figs 1 and 2). It does so despite the initialization data's poor representation of the vortex, which was too big with lower intensities than observed. This problem persists throughout the CTRL run. The atmospheric/mixed layer ocean model was further used to force a storm surge model. Model results were found to compare well with historical observations of storm surge extent and maxima (Fig 3). A set of simulations with uniform SST anomalies applied over the whole region ranging from -4 C and 4 C show the influences of SST on TC characteristics.

The simulations indicate that an increase (decrease) in SST results in an increase (decrease) in intensity, radius of gale-force winds, IKE and precipitation (Fig 4b,c and 5). However, the track prior to landfall was not affected by SST changes (Fig 4a). After landfall the higher intensity storms associated with warmer SST's have a tendency to move further south. Cooler upper ocean temperatures result in an increase in the radius of maximum winds,




although the high wind speeds do not extend to as large radii; this is consistent with an increase in the asymmetry of the storm. In general, changes in TC properties are larger (smaller) for warmer (cooler) SST's. This may be due to there being a limit by how much the values can decrease whilst still maintaining a circulation. On the contrary changes in maximum storm surge are strongest for SST-1 and SST-2 but increases in SST or further decreases have

only a little effect. However, when the storm surge area is considered, an increase in SST results in consistent increases to the total area affected by the storm surge, qualitatively similar to the increases in IKE. Decreases in SST result in large decreases in both maximum storm surge and area affected, with largest decreases occurring between the CTRL and SST-1 and SST-1 and SST-2 simulations. Analysis of the rainfall rates shows that as SST increases the amount of precipitation increases and there is a change in the distribution of the precipitation. Both a

shift from the front-left to front-right quadrants and to larger radii is evident.

As with previous SST sensitivity experiments (e.g. Evans et al. 2004, Kilic and Rable 2013), this is a highly idealised sensitivity study and there are a number of limitations. As Emanuel and Sobel (2013) point out, changing the SST without changing the surface variables that would cause this change will result in imbalances in the surface energy balance and the observed changes may not be realistic. For example, the stability in the lower atmosphere

will have altered which will affect the development of convection and ultimately the intensification of the tropical cyclones and associated rainfall. Therefore, these results are not suitable for quantitative climate change assessments since other changes in the environment important for TC formation and development will also change. However, this study provides qualitative insight into the influence of SST alone on the intensity and characteristics of a tropical cyclone and the associated storm surge. For example, the results suggest maximum storm surge heights

would have been several meters less has a similar TC formed when overall SST were 1-2°C lower (e.g. during non-La Niña conditions). Similarly the overall damage caused by gale-force winds and heavy rainfall would also have been less.

**Data availability**

ERA-Interim re-analysis data are freely available from ECMWF (http://apps.ecmwf.int/datasets/data/interim-full-

daily/). NCEP/MMAB daily sea surface temperature data are freely available from NOAA (http://polar.ncep.noaa.gov/sst/). IBTrACS data is freely available from NOAA (https://www.ncdc.noaa.gov/ibtracs/). Data from the model simulations can be made available on request to the author.




**Acknowledgements**

This work was partially supported by the Australian Climate Change Science Program, funded jointly by the Department of the Environment, the Bureau of Meteorology and CSIRO. Both SL and RH are currently supported through funding from the Australian Government's National Environmental Science Programme. We thank Frank

Colberg, Marcus Thatcher and Paul Branson for their comments that helped to improve the manuscript.

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



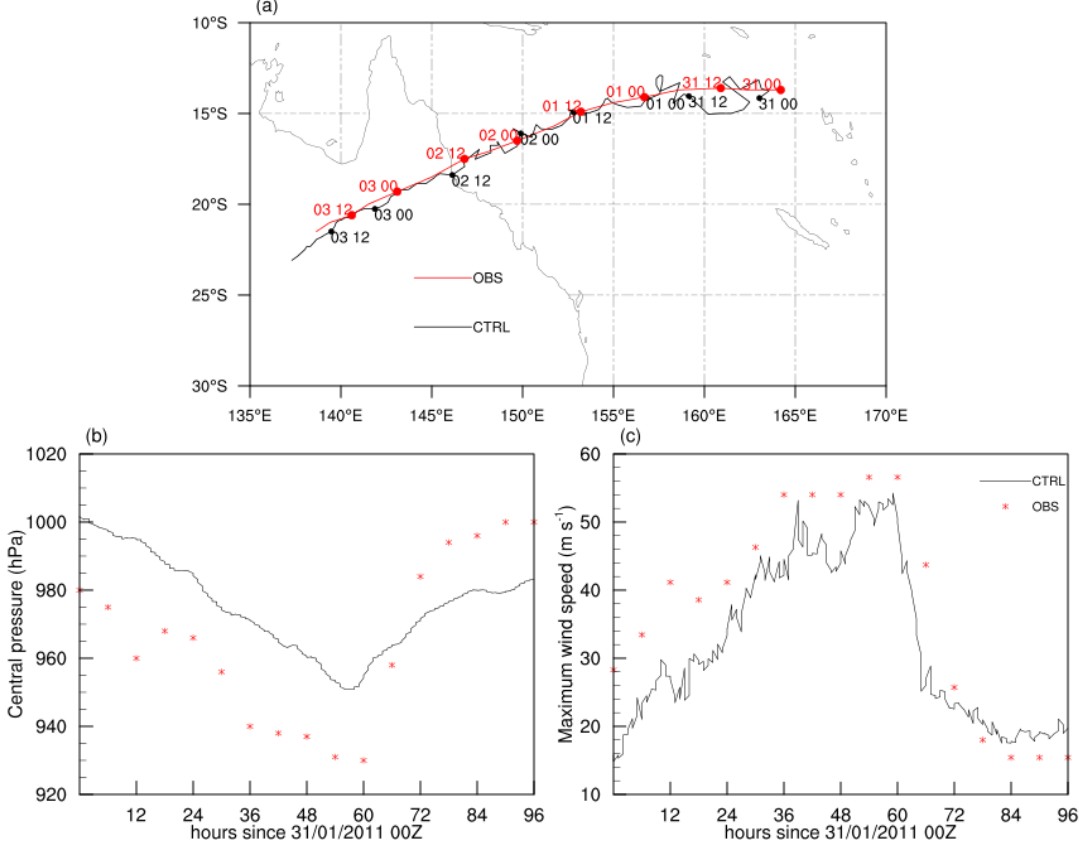

**Figure 1: (a) Simulated (black) and IBTrACS observed (red) track of TC Yasi. Time is indicated. (b) Minimum pressure [hPa] and (c) Maximum windspeed [m s⁻¹] of modelled CTRL simulation of TC Yasi (black line) and observations from IBTrACS (red markers).**





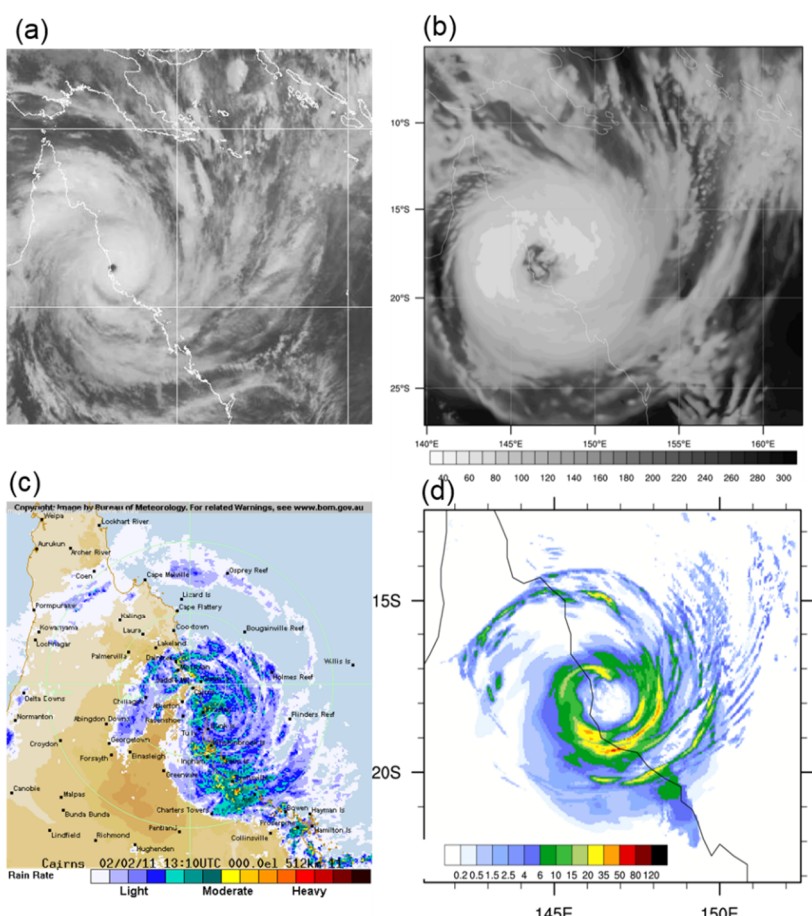

**Figure 2: (a) Satellite IR imagery, (b) modelled (grid 2; 12km resolution) OLR [W m⁻²], (c) radar precipitation (from BoM Cairns radar) and (d) modelled (grid 3; 4 km resolution) 30-minute precipitation rate [mm hr⁻¹] shortly before TC Yasi made landfall.**




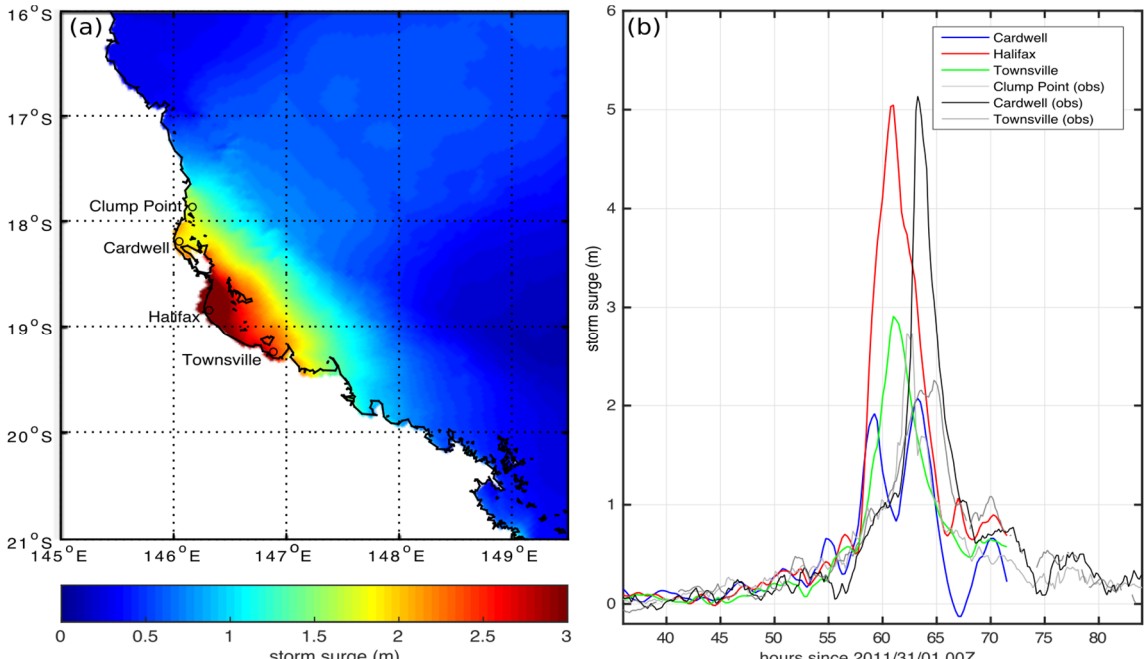

**Figure 3: (a) Maximum simulated storm surge over the CTRL simulation. Open circles indicate location of tide gauge observations and simulation output locations, respectively. (b) Simulated and observed storm surge levels at locations plotted in (a).**

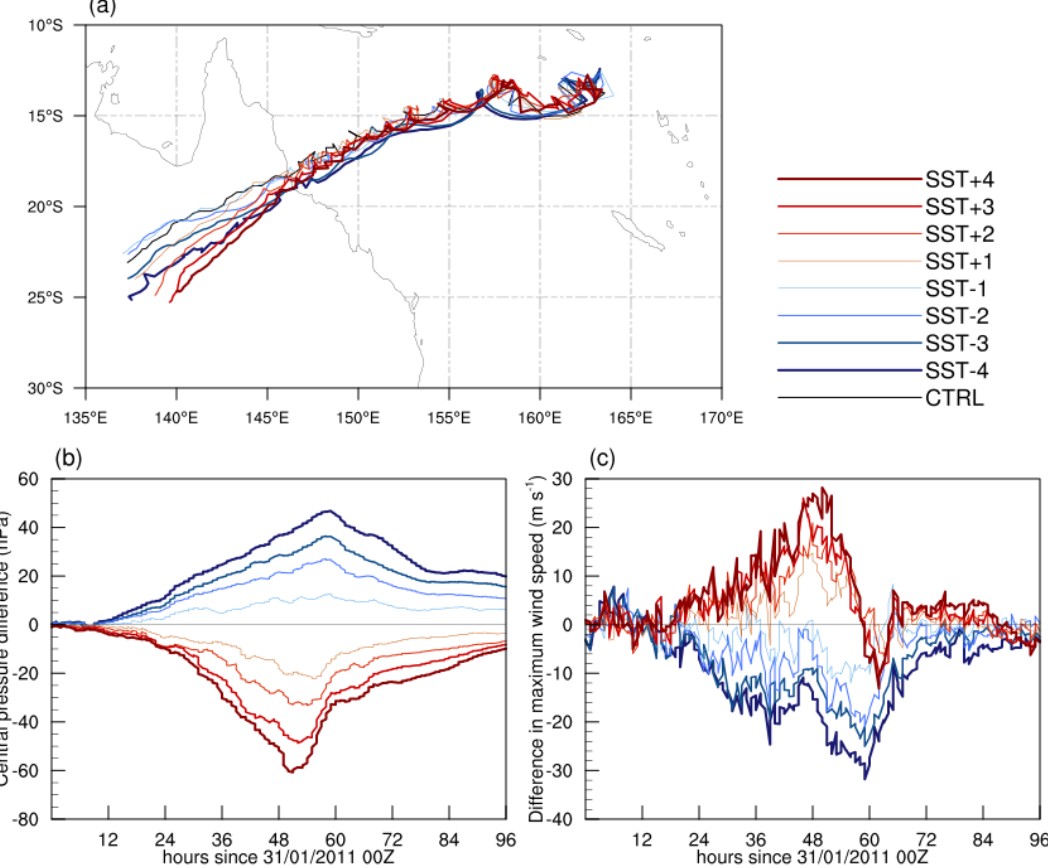

**Figure 4: (a) Tracks of TC Yasi from all nine simulations and timeseries showing the difference (SST runs minus CTRL) in (b) maximum windspeed [m s-1] and (c) minimum pressure [hPa].**





**Figure 5: (a) Percentage difference between radius of maximum winds in SST experiments and CTRL run. (b) Radius to gale-force winds [m] and (c) Integrated Kinetic Energy [TJ] at windspeeds > 17.5 m s⁻¹ for all model simulations.**




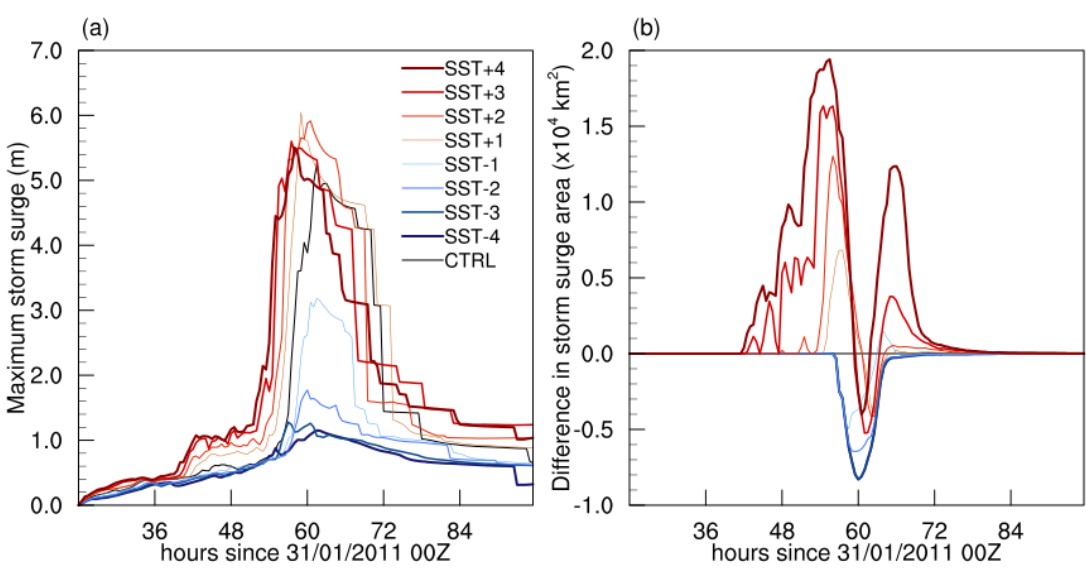

**Figure 6: (a) Maximum simulated storm surge (m) for all runs. (b) Difference in Storm surge area (defined as area of water levels > 1 m in km$^2$) for each SST simulation minus the CTRL.**

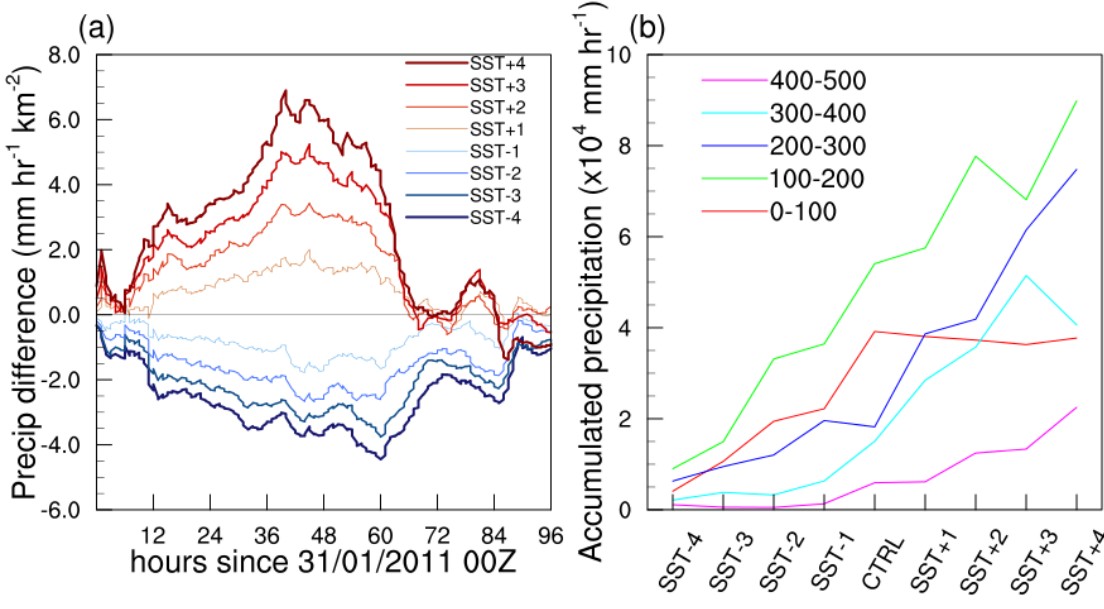

**Figure 7: (a) Difference in precipitation rate within 500 km of the storm centre for each SST simulation minus the CTRL. (b) Accumulated precipitation (mm hr-1; total over all grid points) within different radii for each simulation at landfall**



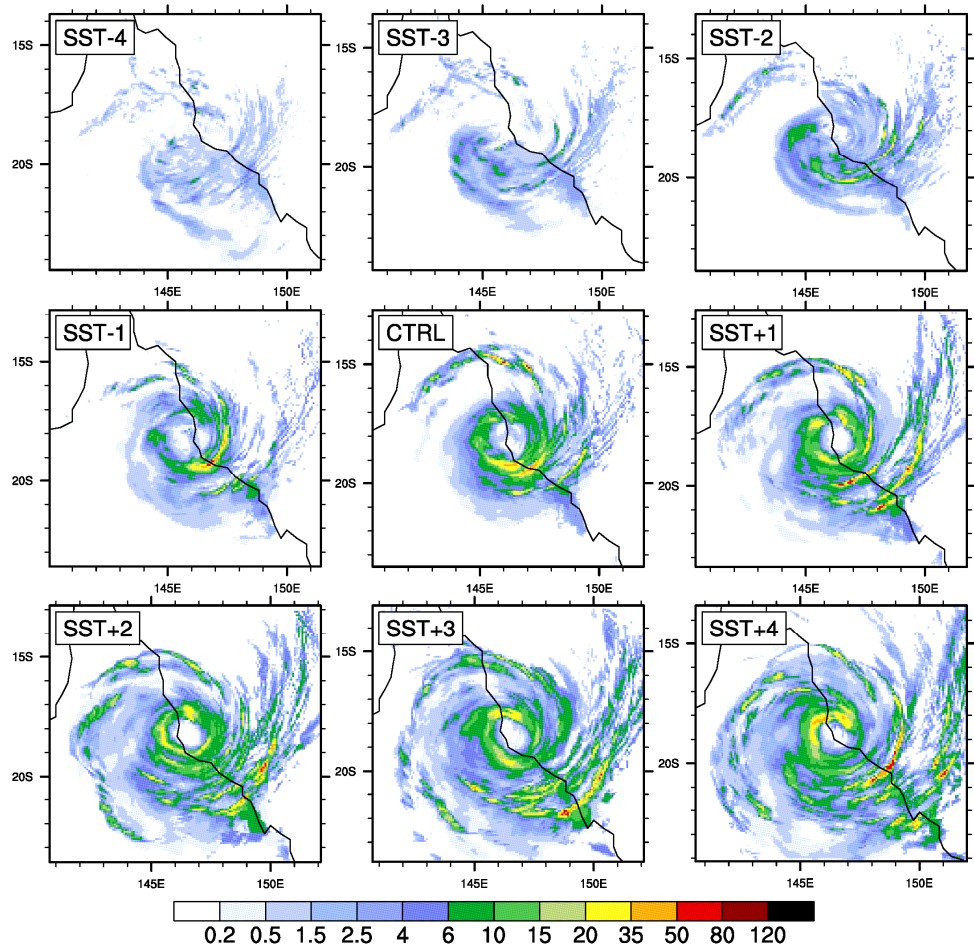

**Figure 8: 30 minute precipitation rate [mm hr$^{-1}$] as the storm makes landfall for each of the nine simulations**