# Peer review of "The influence of sea surface temperature on the intensity and associated storm surge of tropical cyclone Yasi: A sensitivity study"

_Natural Hazards and Earth System Sciences, 2017_

## Referee Comment (RC1) · C. Arthur (Referee) · 22 Dec 2017

Abstract should be significantly shortened and focus on key outcomes of the study (see 6th point below). It is not immediately clear from the abstract what the focus of the study is - the ability of the model to simulate TC Yasi, or influence of SST changes on storm morphology or the resulting storm surge heights.

There is no further mention in the manuscript of impacts or mitigation - suggest removing this from the abstract.

Page 3, line 5: in-text reference to Miglietta (2011) should be Miglietta et al. (2011)

Page 8, lines 11-14: The sensitivity of maximum storm surge height to TC forward speed, location of landfall and RMW would be useful to open the discussion on the sensitivity to SST. I suggest reworking this paragraph and the preceding one to indicate the above characteristics influence storm surge height as background to the discussion on variability with SST (i.e. move these lines ahead of the preceding paragraph to help set the context of the discussion on SST influence).

Page 10, line 11: in-text reference to Evans et al. (2004) - should this be Evans et al. (1994), or is there a reference missing in the reference list?

I would like to see more discussion (throughout if possible) on the final point made in the manuscript: "...the results suggest maximum storm surge heights would have been several metres less had a similar TC formed when overall SST were 1-2C lower...". This is probably the most important outcome of the sensitivity study, but it is only included as the second to last sentence. In the context of future climate scenarios, increasing storm surge heights (and precipitation) associated with increasing SSTs is a major finding and deserves more (and more prominent) discussion.

Figure 4: caption for subplots (b) and (c) need to be swapped.

Figure 5: all panels need a label on the horizontal axis.

---

## Referee Comment (RC2) · Anonymous Referee #2 · 8 Jan 2018

The term 'sensitivity study' is a little confusing. A more suitable term might be 'idealized study' since this study examines the response of TC Yasi to warmer/colder SSTs in an idealized environment.

The term 'potential destructiveness' should be clearly defined in the manuscript. There are a number of factors that can determine the 'potential destructiveness' of a tropical cyclone. These include, but are not limited to, the characteristics of the tropical cyclone (e.g. max winds, size, translation speed) and the populated areas affected by the TC (e.g. population size, infrastructure). As such, it would add clarity to the manuscript if the term 'potential destructiveness' was defined.

---

## Author Comment (AC1) · 30 Jan 2018

RC2, comment 1: The term 'sensitivity study' is a little confusing. A more suitable term might be 'idealized study' since this study examines the response of TC Yasi to warmer/colder SSTs in an idealized environment.

Response: We would disagree, a sensitivity study is the response of small changes in an input variable and hence the response to TC Yasi to warmer/cooler SSTs is just that. The term idealised is included throughout the paper.

To further clarify this line 11 (in the abstract) will be altered in the revised version: "In

this study, a conceptually simple idealised sensitivity analysis...."

RC2, comment 2: The term 'potential destructiveness' should be clearly defined in the manuscript. There are a number of factors that can determine the 'potential destructiveness' of a tropical cyclone. These include, but are not limited to, the characteristics of the tropical cyclone (e.g. max winds, size, translation speed) and the populated areas affected by the TC (e.g. population size, infrastructure). As such, it would add clarity to the manuscript if the term 'potential destructiveness' was defined.

Response: In the manuscript there is already the line (pg 7, line 26): "The integrated kinetic energy (IKE, Powell & Reinhold 2007) takes into account both maximum wind speeds and storm size and is therefore a good measure of the destructiveness of a TC."

The main text and abstract will be altered in the revised version to further clarify this.

---

## Author Comment (AC2) · 30 Jan 2018

The authors thank Craig for his constructive comments.

RC1, comment 1: Abstract should be significantly shortened and focus on key outcomes of the study (see 6th point below). It is not immediately clear from the abstract what the focus of the study is - the ability of the model to simulate TC Yasi, or influence of SST changes on storm morphology or the resulting storm surge heights.

Response: A possible revised abstract:

"Tropical cyclones (TCs) cause widespread damage associated with strong winds,

heavy rainfall and storm surge. TC Yasi was one of the most powerful TCs to im-pact the Queensland coast since records began. Prior to Yasi, the SSTs in the Coral Sea were higher than average by 1-2°C, primarily due to the 2010/2011 La Niña event. In this study, a conceptually simple idealised sensitivity analysis is performed using a high resolution regional model to gain insight into the influence of SST on the track, size intensity and associated rainfall of TC Yasi. A set of nine simulations with uniform SST anomalies of between -4°C and 4°C applied to the observed SST's are carried out. The resulting surface winds and pressure are used to force a barotropic storm surge model to examine the influence of SST on the associated storm surge of TC Yasi. An increase in SST results in an increase in intensity, precipitation and integrated kinetic energy of the storm, however there is little influence on track prior to landfall. In addition to an increase in precipitation, there is a change in the spatial distribution of precipitation as the SST increases. Decreases in SSTs result in an increase in the radius of maximum winds due to an increase in the asymmetry of the storm, although the radius of gale-force winds decreases. These changes in the TC characteristics also lead to changes in the associated storm surge. Generally, cooler (warmer) SST lead to reduced (enhanced) maximum storm surges. However, the increase in surge reaches a maximum with an increase in SST of 2 °C. Any further increase in SST does not affect the maximum surge but the total area and duration of the simulated surge increases with increasing upper ocean temperatures. A large decrease in maximum storm surge height occurs when a negative SST anomaly is applied, suggesting if TC Yasi had occurred during non-La Niña conditions the associated storm surge would have been greatly diminished, with a decrease in storm surge height of over 3m when the SST is reduced by 2 °C. In summary, increases in SST lead to an increase in the potential destructiveness of TCs with regards to intensity, precipitation and storm surge, although this relationship is not linear."

RC1, comment 2: There is no further mention in the manuscript of impacts or mitigation - suggest removing this from the abstract.

Response: This line will be removed from the abstract.

RC1, comment 3: Page 3, line 5: in-text reference to Miglietta (2011) should be Miglietta et al. (2011).

Response: Thanks for spotting this, altered.

RC1, comment 4: Page 8, lines 11-14: The sensitivity of maximum storm surge height to TC forward speed, location of landfall and RMW would be useful to open the discussion on the sensitivity to SST. I suggest reworking this paragraph and the preceding one to indicate the above characteristics influence storm surge height as background to the discussion on variability with SST (i.e. move these lines ahead of the preceding paragraph to help set the context of the discussion on SST influence).

Response: Moved and reworked in revised version.

RC1, comment 5: Page 10, line 11: in-text reference to Evans et al. (2004) - should this be Evans et al. (1994), or is there a reference missing in the reference list?

Response: Yes, it should be 1994, thanks for spotting. Changed accordingly

RC1, comment 6: I would like to see more discussion (throughout if possible) on the final point made in the manuscript: "...the results suggest maximum storm surge heights would have been several metres less had a similar TC formed when overall SST were 1-2C lower...". This is probably the most important outcome of the sensitivity study, but it is only included as the second to last sentence. In the context of future climate scenarios, increasing storm surge heights (and precipitation) associated with increasing SSTs is a major finding and deserves more (and more prominent) discussion.

Response: Agreed. This is discussed further in the discussion of Fig 6 (pg 8) and further emphasised in the reworked abstract.

RC1, comment 7: Figure 4: caption for subplots (b) and (c) need to be swapped.

Response: Thanks for spotting, altered.

RC1, comment 8: Figure 5: all panels need a label on the horizontal axis.

Response: Label added to x-axis: "hours since 31/01/2011 00Z"
* * *

---

## Author Response (AR1)

**Reviewer 1**

The authors thank Craig for his constructive comments.

5    1) Abstract should be significantly shortened and focus on key outcomes of the study (see 6th point below). It is not immediately clear from the abstract what the focus of the study is - the ability of the model to simulate TC Yasi, or influence of SST changes on storm morphology or the resulting storm surge heights.

The abstract has been revised:

10    "Tropical cyclones (TCs) result in widespread damage associated with strong winds, heavy rainfall and storm surge. TC Yasi was one of the most powerful TCs to impact the Queensland coast since records began. Prior to Yasi, the SSTs in the Coral Sea were higher than average by 1-2°C, primarily due to the 2010/2011 La Niña event. In this study, a conceptually simple idealised sensitivity analysis is performed using a high resolution regional model to gain insight into the influence of SST on the track, size, intensity and associated rainfall of TC Yasi. A set of nine simulations with uniform SST anomalies of between

15    -4°C and 4°C applied to the observed SST's are analysed. The resulting surface winds and pressure are used to force a barotropic storm surge model to examine the influence of SST on the associated storm surge of TC Yasi.

An increase in SST results in an increase in intensity, precipitation and integrated kinetic energy of the storm, however there is little influence on track prior to landfall. In addition to an increase in precipitation, there is a change in the spatial distribution of precipitation as the SST increases. Decreases in SSTs result in an increase in the radius of maximum winds due to an

20    increase in the asymmetry of the storm, although the radius of gale-force winds decreases. These changes in the TC characteristics also lead to changes in the associated storm surge. Generally, cooler (warmer) SST lead to reduced (enhanced) maximum storm surges. However, the increase in surge reaches a maximum with an increase in SST of 2 °C. Any further increase in SST does not affect the maximum surge but the total area and duration of the simulated surge increases with increasing upper ocean temperatures. A large decrease in maximum storm surge height occurs when a negative SST anomaly

25    is applied, suggesting if TC Yasi had occurred during non-La Niña conditions the associated storm surge may have been greatly diminished, with a decrease in storm surge height of over 3m when the SST is reduced by 2 °C.

In summary, increases in SST lead to an increase in the potential destructiveness of TCs with regards to intensity, precipitation and storm surge, although this relationship is not linear."

30    2) There is no further mention in the manuscript of impacts or mitigation - suggest removing this from the abstract.

This line has been removed from the abstract.

3) Page 3, line 5: in-text reference to Miglietta (2011) should be Miglietta et al. (2011).

Thanks for spotting this, altered.

4) Page 8, lines 11-14: The sensitivity of maximum storm surge height to TC forward speed, location of landfall and RMW would be useful to open the discussion on the sensitivity to SST. I suggest reworking this paragraph and the preceding one to indicate the above characteristics influence storm surge height as background to the discussion on variability with SST (i.e. move these lines ahead of the preceding paragraph to help set the context of the discussion on SST influence).

Moved and reworked in revised version (page 7, lines 19-28)

5) Page 10, line 11: in-text reference to Evans et al. (2004) - should this be Evans et al. (1994), or is there a reference missing in the reference list?

Yes, it should be 1994, thanks for spotting. Changed accordingly

6) I would like to see more discussion (throughout if possible) on the final point made in the manuscript: "...the results suggest maximum storm surge heights would have been several metres less had a similar TC formed when overall SST were 1-2C lower...". This is probably the most important outcome of the sensitivity study, but it is only included as the second to last sentence. In the context of future climate scenarios, increasing storm surge heights (and precipitation) associated with increasing SSTs is a major finding and deserves more (and more prominent) discussion.

Agreed. This is discussed further in the discussion of Fig 6 (pg 7) and further emphasised in the reworked abstract.

7) Figure 4: caption for subplots (b) and (c) need to be swapped.

Thanks for spotting, swapped.

8) Figure 5: all panels need a label on the horizontal axis.

Label added to x-axis: "hours since 31/01/2011 00Z"

**Reviewer 2**

1) The term 'sensitivity study' is a little confusing. A more suitable term might be 'idealized study' since this study examines the response of TC Yasi to warmer/colder SSTs in an idealized environment.

5    We would disagree, a sensitivity study is the response of small changes in an input variable and hence the response to TC Yasi to warmer/cooler SSTs is just that.  The term idealised is included throughout the paper.

To further clarify this, the abstract now contains the following:

 "In this study, a conceptually simple **idealised** sensitivity analysis"

10   2) The term 'potential destructiveness' should be clearly defined in the manuscript. There are a number of factors that can determine the 'potential destructiveness' of a tropical cyclone. These include, but are not limited to, the characteristics of the tropical cyclone (e.g. max winds, size, translation speed) and the populated areas affected by the TC (e.g. population size, infrastructure). As such, it would add clarity to the manuscript if the term 'potential destructiveness' was defined.

In the manuscript there is already the line (pg 7, line 26): "The integrated kinetic energy (IKE, Powell & Reinhold 2007) takes

15   into account both maximum wind speeds and storm size and is therefore a good measure of the destructiveness of a TC."

The text has been altered so that any reference to potential destructiveness includes more detail as to what this means.

[revised manuscript text omitted]